# Correlates of behavioral and emotional disorders among school-going adolescents in Uganda

Max Bobholz[1], Julia Dickson-Gomez[1], Catherine Abbo[2]*, Arthur Kiconco[1], Abdul R. Shour[3], Simon Kasasa[4], Laura D. Cassidy[1], Ronald Anguzu[1]

**1** Institute for Health and Humanity, Medical College of Wisconsin, Milwaukee, Wisconsin, United States of America, **2** Makerere University College of Health Sciences, Kampala, Uganda, **3** Essentia Institute of Rural Health, Research Operations, Duluth, Minnesota, United States of America, **4** Department of Epidemiology and Biostatistics, Makerere University School of Public Health, Kampala, Uganda

* cathya180@gmail.com, cathyabbo@chs.mak.ac.ug

## Abstract

### Background

Adolescence is a critical development transition period that increases vulnerability to poor mental health outcomes. Recent evidence suggests that adolescents in Uganda experience high rates of behavioral and emotional disorders. We examined the factors associated with emotional and behavioral health outcomes among school-going adolescents in Uganda.

### Methods

In this cross-sectional study, we surveyed 1,953 students aged 10–18 enrolled in Central and Eastern Uganda secondary schools selected by stratified random sampling. Our outcome variables were (i) emotional and (ii) behavioral disorders that were measured using the Child and Adolescent Symptom Inventory-5 (CASI-5) diagnostic criteria outlined in the Diagnostic Statistical Manual-5 (DSM-5). Emotional disorders included major depressive disorder, generalized anxiety disorder, social anxiety disorder, and separation anxiety disorder. Attention deficit/hyperactivity disorder, conduct disorder, and oppositional defiant disorder were considered behavioral disorders. Covariates included socio-demographic, hardship-related experiences, and school-related characteristics. Modified Poisson and logistic regression models were appropriately run for the factors independently associated with respective outcomes. Prevalence ratios (PR), odds ratios (OR), and corresponding 95% confidence intervals (95%CI) were reported with $p < 0.05$ considered significant.

### Results

Participants' mean age was 15.5 (SD = 2.0) years; 54.7% were female, 5.7% had a behavioral disorder, and 17.4% had an emotional disorder. In the adjusted models,

**Data availability statement:** All relevant data are within the paper and its Supporting Information files.

**Funding:** CA received funding from Styrelsen för Internationellt Utvecklingssamarbete (SIDA/SAREC; No: 51180060) to implement this study. No additional external funding was received for this study. The funder had no role in study design, data collection and analysis, decision to publish, or preparation of the manuscript.

**Competing interests:** The authors have declared that no competing interests exist.

factors independently associated with higher odds of behavioral disorder were age (OR=1.2; 95%CI 1.1 - 1.4) and family history of mental illness (OR=1.9; 95%CI 1.2 - 3.3). Factors independently associated with a higher risk of emotional disorder were being female (PR = 1.5; 95%CI 1.2 - 1.8), being enrolled in advanced education (PR = 1.7; 95%CI 1.2 - 2.4) and attending private school (PR = 1.4; 95%CI 1.1 - 1.8).

## Conclusions

Behavioral and emotional disorders are prevalent among adolescents enrolled in secondary schools in Central and Eastern Uganda. Investigating potential causal pathways of the identified associations is critical to support school mental health initiatives. School-based programs should enhance routine mental health assessments and target at-risk students in order to improve the mental health of school-going adolescents in Uganda.

## Background

Mental health conditions are a significant public health problem and are ranked the sixth leading cause of health loss and disability globally [1]. An estimated 13.9% of the global population experiences mental disorders, with depression and anxiety disorders being the most common in all age groups [1]. In adolescents (ages 10–19 years), one in seven experiences a mental disorder, which accounts for 13% of the global burden of disease in adolescents [1]. Mental disorders, especially depression, have risen considerably in low, and middle-income countries (LMICs) since 1990 [1]. Depression, anxiety, and behavioral disorders are among the leading causes of illness and disability among adolescents, and suicide is the fourth leading cause of death among individuals aged 15–29 years [2]. Depression alone accounts for 35% of the global burden of disease or disease-adjusted life years (DALYs), in addition to 700,000 annual deaths due to suicide [3,4]. In Uganda, estimates show the prevalence of depression among children and adolescents at 23.6% [4]. Mental health conditions can result from or lead to life difficulties, including poor relationships with family, friends, community, and problems at school and work. Adolescence is a critical transition period during which biological and social development occurs [5]. Developmental disruptions during this time can cause lasting developmental and health impacts on individuals [6–8]. Since a substantial amount of an individual's lifetime is spent in school settings during this period, the school environment plays a critical role in their social and emotional development. A safe and healthy school environment is essential for a child's successful neurodevelopment and ability to thrive [9].

The burden (prevalence) of emotional and behavioral disorders among Uganda adolescents varies according to many factors, such as HIV status [10,11], trauma [12], and poverty [12,13]. Recent studies using the Child and Adolescent Symptom Inventory-5 (CASI-5) among perinatally HIV-infected youth in Uganda have described a prevalence of emotional and behavioral disorders at 11.5% and 9.6%, respectively [14]. In Uganda, Kinyanda and colleagues reported the risk factors for behavioral

disorders as male and older (adolescent vs. child) [14]. Another article utilizing similar methods for surveying symptoms reported a prevalence of ADHD at 6% among HIV-infected youth in Uganda. Few studies, however, have investigated the burden of emotional and behavioral disorders and the factors associated with these respective mental health disorders among school-going adolescents in Uganda [15–20]. Similarly, little investigation has been done to understand the role of experiencing hardship and the individual's school environment in mental health outcomes.

Screening for mental health disorders is an essential preventive mental health strategy aimed at reducing the occurrence and rising burden of mental disorders [21,22]. Effective ways to survey adolescent mental and behavioral health have long been a focus of the scientific community; many test batteries have been developed to assess respondents' burden of these outcomes. The CASI-5, published in 2013, was developed for children aged 5–18 as a behavior rating scale for conditions recognized by the Diagnostic Statistical Manual-5 (DSM-5) [23]. The CASI-5 battery has been cross-culturally adapted for use in parts of Uganda [24]. Ensuring that mental health screening tools are able to appropriately capture the burden of mental health disorders in diverse populations is essential.

Beyond understanding how to screen for mental health conditions, it is also critical to address the conditions in which adolescents are experiencing mental health challenges. Mental health in school settings should be a crucial focus of school administrators, public health practitioners, and policymakers. A recent systematic review showed that school-based mental health interventions can effectively improve mental health literacy and reduce stigma [25]. Coordinated comprehensive efforts of stakeholders such as school teachers and counselors are critical in addressing student mental health [26,27]. Therefore, understanding the burden of emotional and behavioral health outcomes in secondary schools should inform stakeholder strategies to promote adolescent mental health and increase undetected mental health disorders while reducing their long-term negative health impact.

Although the burden of emotional and behavioral disorders among HIV-infected Ugandan youth has been explored, little investigation has been done to understand the role of experiencing hardship (specifically living in less-permanent housing, being exposed to domestic violence, having a family history of mental illness, and being orphaned) and the individual's school environment in mental health outcomes. In the current study, we aimed to assess the factors associated with mental and behavioral health disorders among adolescents enrolled in secondary schools in central and eastern Uganda. We hypothesized that hardship experiences, demographic characteristics, and school features contribute to and could be associated with the prevalence of emotional and behavioral disorders among school-going adolescents in Uganda.

## Methods

### Study design, setting, and population

We used a cross-sectional study design to survey 1,972 adolescents aged 12–18 years enrolled in eight secondary schools in Iganga district in Eastern Uganda and Mukono district in Central Uganda. Data was collected between June 2017 and November 2017. All respondents provided informed consent to participate in the study; consent was obtained from the parents or guardians of participants under the age of 18 years.

### Sampling strategy

We purposively selected Uganda's two most populous regions, Central and Eastern. Iganga district schools (Eastern region) are predominantly rural, while Mukono district schools (Central region) are predominantly urban. For school sampling, we selected one school district from each region based on population and past academic performance. Using purposive, stratified random sampling, we selected eight secondary schools, four from each district. Each district had two government-funded and two private schools. We pre-visited the District Education Officers (DEO) for the list of all

registered secondary schools in the respective districts. DEOs implement education laws and regulations according to government policies. We then created a list of schools and categorized them by their rural or urban status.

## Data collection and tools

Demographic surveys and CASI-5 questionnaires were administered to the adolescents directly by Psychiatric Clinical Officers (PCOs) who are healthcare providers trained and familiar with mental health diagnoses. The battery consists of more than one hundred questions about the frequency of a symptom's presence, surveying 24 emotional and behavioral disorders. Each section of CASI-5 asks respondents about the symptoms of specific disorders. At the end of each section, a question is asked assessing how often the individual feels their symptoms impair social or behavioral function. Respondents were excluded from this analysis if they responded to less than half of the questions in at least one sub-section corresponding to emotional or behavioral disorders included in this analysis. For example, individuals who did not answer any questions on ADHD symptoms but completed all other pertinent questions were removed from the analysis. Less than 1% (n = 19) of the study population was excluded, resulting in 1,953 remaining participants.

## Study measures

**Dependent variables.** The CASI-5 battery was used to screen selected study participants for the presence of symptoms of various mental health conditions. Responses to the CASI-5 survey are quantified by a Likert scale: 0 = Never, 1 = Sometimes, 2 = Often, 3 = Very Often. For this study, a response of at least 2 (Often or Very Often) indicated that the participant had the symptom. The DSM-5 guidelines were used to determine the presence of an emotional or behavioral disorder [28]. An alternative scoring method (t-score) was used to confirm our rates of major depressive disorder in the study population [23]. Our composite categorical outcome variables (yes/no) were (i) the presence of an emotional disorder and (ii) the presence of a behavioral disorder. Emotional disorders used in this analysis include major depressive disorder, generalized anxiety disorder, social anxiety disorder, and separation anxiety disorder. Attention deficit/hyperactivity disorder, conduct disorder, and oppositional defiant disorder were included in behavioral disorder analyses. Respondents who had emotional and behavioral problems were referred to the nearest health facility for further evaluation and management.

**Covariates.** A demographic questionnaire was also administered alongside the CASI-5 battery. Demographic variables included sex (female/male), age (continuous), urbanity (rural/urban), and level of education [Advanced (A') or Ordinary (O') level]. Hardship variables were the nature of housing (permanent/ semi-permanent/hut), presence of domestic violence in the home (yes/no), family history of mental illness (yes/no), and orphanhood (at least one parental death/both parents living). School characteristics included sex status (mixed boys and girls/single sex) and school ownership (private/government). No manipulation was required for responses to the demographic survey as data did not require cleaning or recoding.

## Ethics approval and consent to participate

Ethical approval was obtained from the School of Medicine Research Ethics Committee (SOMREC). Permission to conduct the study was also obtained from the Uganda National Council for Science and Technology (UNCST). All respondents provided informed consent to participate in the study; consent was obtained from the parents or guardians of participants under the age of 18 years.

## Statistical analysis

Descriptive statistics were obtained by computing frequencies and corresponding proportions of respondents who responded to the question and fit the emotional or behavioral disorder diagnostic criteria. In bivariate analysis, a

chi-squared test of association was conducted between each categorical demographic characteristic and the presence of an emotional/behavioral disorder. Two-sample t-tests were used to assess mean age differences between each group. Further, bivariate logistic regression analyses were conducted for the unadjusted associations between independent variables and behavioral disorders. Modified Poisson regression models [29] were used for analyzing associations between independent variables and the presence of an emotional disorder. Multivariable regression analyses were performed to understand the nature of associations while controlling for hardship experiences, demographic characteristics, and school features. We fitted the adjusted models with variables that were statistically significant at bivariate analysis and variables that were biologically plausible or potential confounders. Prevalence ratios (PR), and odds ratios (OR), were reported for logistic regression and modified Poisson regression models respectively, and their corresponding 95% confidence intervals (95%CI). The alpha threshold for statistical significance was set at 0.05.

### Ethics approval and consent to participate

Ethical approval was obtained from the School of Medicine Research Ethics Committee (SOMREC). Permission to conduct the study was also obtained from the Uganda National Council for Science and Technology (UNCST). All respondents provided informed consent to participate in the study; consent was obtained from the parents or guardians of participants under the age of 18 years.

## Results

### Participant demographics

Overall, among the study population of 1,953 school-going adolescents, their mean age was 15.47 years [standard deviation (SD)=1.99], and more were female (57.55%, n = 1,124), urban dwellers (52.38%; n = 1,023) and 12.35% (n = 236) were enrolled in A' level education. An overall proportion of 17.36% (n = 340) reported symptoms suggestive of an emotional disorder, and 5.71% (n = 112) had behavioral disorders. Few participants (n = 55, 2.78%) were considered to have both an emotional and behavioral disorder. Most respondents (59.04%; n = 1,153) attended private schools, and 89.71% (n = 1,752) studied in mixed-gender schools. In terms of hardship experiences of respondents, 14.03% (n = 263) reported witnessing domestic violence in the home setting, 18.82% had a family history of mental illness, and 17.10% (n = 332) were single or two-parent orphans. Few respondents (1.48%; n = 28) reported living in hut-style homes, 82.40% (n = 1,559) lived in permanent residences, and 16.12% of respondents (n = 305) lived in semi-permanent homes (Table 1).

### Prevalence and correlates of emotional disorders

The prevalence of emotional disorders ranged from 0.20% (major depressive disorder) to 14.20% (separation anxiety disorder). Generalized anxiety disorder (3.06%) and social anxiety disorder (2.60%) were also observed in the study population. Low depression prevalence was reinforced by applying the symptom severity T-score approach (23), observing that only mild depression was present and at very low levels [30]; no individuals met the threshold for moderate or severe depression with this method. The prevalence of individual emotional and behavioral disorders can be found in S1 Table. In the bivariate regression analyses, females (PR = 1.48; 95%CI 1.20 - 1.82) and attending private school (PR = 1.41; 95%CI 1.16 - 1.71) had 1.5 and 1.4 times the prevalence of emotional disorders when compared to male students and attending public schools, respectively. In other significant bivariate regression analyses, the prevalence of emotional disorders was significantly higher among urban-dwelling students (PR = 1.33; 95%CI 1.09 - 1.62), those living in semi-permanent homes (PR = 1.35; 95%CI 1.06 - 1.71), those who witnessed domestic violence (PR = 1.33; 95%CI 1.03 - 1.71), and those who had a family history of mental illness (PR = 1.32; 95%CI 1.05 - 1.67). In the adjusted regression model, factors independently associated with the presence of an emotional disorder were being female (adj. PR = 1.46; 95%CI 1.15 - 1.84),

PLOS Global Public Health

**Table 1. Bivariate tests of simple association (chi-squared) among demographic characteristics, hardship experiences, school features, and the presence of emotional or behavioral disorders among Ugandan school-going adolescents.**

| Characteristic | Totals n (%) | Emotional disorder | | | Behavioral disorder | | |
|---|---|---|---|---|---|---|---|
| | | Yes n (%) | No n (%) | χ2 | Yes n (%) | No n (%) | χ2 |
| **Sex** | | | | | | | |
| Male | 829 (42.45) | 113 (13.61) | 717 (86.39) | 14.121*** | 48 (5.76) | 785 (94.24) | 0.006 |
| Female | 1,124 (57.55) | 227 (20.12) | 901 (79.88) | | 64 (5.68) | 1,063 (94.32) | |
| **Age [mean (sd)]** † | 15.472 (1.99) | 15.400 | 15.486 | p=0.224 | 15.938 | 15.441 | p=0.005** |
| **Urbanity** | | | | | | | |
| Rural | 930 (47.62) | 138 (14.81) | 794 (85.19) | 8.109** | 47 (5.03) | 888 (94.97) | 1.569 |
| Urban | 1,023 (52.38) | 202 (19.69) | 824 (80.31) | | 65 (6.34) | 960 (93.66) | |
| **Level of education enrolled in** | | | | | | | |
| Advanced level | 236 (12.35) | 49 (20.76) | 187 (79.24) | 2.005 | 14 (5.91) | 223 (94.09) | 0.015 |
| Ordinary level | 1,675 (87.65) | 286 (17.02) | 1,394 (82.98) | | 96 (5.71) | 1,585 (94.29) | |
| **Nature of housing** | | | | | | | |
| Permanent | 1,559 (82.40) | 257 (16.45) | 1,305 (83.55) | 8.136* | 85 (5.43) | 1,481 (94.57) | 4.108 |
| Semi-permanent | 305 (16.12) | 68 (22.15) | 239 (77.85) | | 18 (5.90) | 287 (94.10) | |
| Hut | 28 (1.48) | 8 (28.57) | 20 (71.43) | | 4 (14.29) | 24 (85.71) | |
| **Domestic violence witness** | | | | | | | |
| Yes | 263 (14.03) | 58 (22.05) | 205 (77.95) | 4.738* | 21 (7.95) | 243 (92.05) | 3.106 |
| No | 1,612 (85.97) | 268 (16.57) | 1,349 (83.43) | | 85 (5.26) | 1,532 (94.74) | |
| **Family history of mental illness** | | | | | | | |
| Yes | 328 (18.82) | 73 (22.26) | 255 (77.74) | 5.322* | 31 (9.42) | 298 (90.58) | 11.152*** |
| No | 1,415 (81.18) | 239 (16.84) | 1,180 (83.16) | | 67 (4.72) | 1,352 (95.28) | |
| **Orphan** | | | | | | | |
| Yes | 332 (17.10) | 62 (18.62) | 271 (81.38) | 0.401 | 16 (4.78) | 319 (95.22) | 0.703 |
| No | 1,610 (82.90) | 277 (17.17) | 1,336 (82.83) | | 96 (5.95) | 1,518 (94.05) | |
| **School sex status** | | | | | | | |
| Mixed | 1,752 (89.71) | 310 (17.64) | 1,447 (82.36) | 0.929 | 101 (5.75) | 1,657 (94.25) | 0.030 |
| Single-sex | 201 (10.29) | 30 (14.93) | 171 (85.07) | | 11 (5.45) | 191 (94.55) | |
| **School ownership** | | | | | | | |
| Private | 1,153 (59.04) | 168 (20.95) | 984 (79.05) | 12.153*** | 45 (5.62) | 755 (94.38) | 0.020 |
| Government | 800 (40.96) | 172 (14.88) | 634 (85.12) | | 67 (5.78) | 1,093 (94.22) | |
| **Total** | 1,953 (100) | 340 (17.36) | 1,618 (82.64) | -- | 112 (5.71) | 1,848 (94.29) | -- |

†: a t-test was performed to assess differences in mean age

*<0.05;

**<0.01;

***<0.001

compared to male students, and attending private school (adj. PR = 1.40; 95%CI 1.12 - 1.76), compared to attending public school (Table 2; Fig 1).

## Prevalence and correlates of behavioral disorders

The prevalence of the behavioral disorders ranged from 1.89% (conduct disorder) to 2.96% (opposition defiant disorder). The lowest prevalence of emotional disorders was 0.20% for major depressive disorder, with the highest prevalence of 14.20% for separation anxiety disorder. ADHD was observed in 2.35% of respondents. In the unadjusted regression analyses, age

**Table 2. Logistic and modified Poisson regression models for assessing association of demographic characteristics, hardship experiences, school features and the presence of emotional or behavioral disorders among Ugandan school-going adolescents.**

| Characteristic | Emotional disorder | | | | Behavioral Disorder | | | |
|---|---|---|---|---|---|---|---|---|
| | PR | 95% CI | APR | 95% CI | OR | 95% CI | AOR | 95% CI |
| **Sex** | | | | | | | | |
| Male | Ref | -- | Ref | -- | Ref | -- | Ref | -- |
| Female | 1.478*** | 1.20-1.82 | 1.457** | 1.15-1.84 | 0.985 | 0.67-1.45 | 1.156 | 0.72-1.85 |
| **Age [mean (sd)]** | 0.982 | 0.93-1.03 | 0.966 | 0.91-1.03 | 1.126* | 1.03-1.23 | 1.205** | 1.05-1.38 |
| **Urbanity** | | | | | | | | |
| Rural | Ref | -- | Ref | -- | Ref | -- | Ref | -- |
| Urban | 1.330** | 1.09-1.62 | 1.216 | 0.96-1.54 | 1.279 | 0.87-1.88 | 1.224 | 0.76-1.98 |
| **Level of education** enrolled in | | | | | | | | |
| Advanced level | 1.220 | 0.93-1.60 | 1.690** | 1.18-2.42 | 1.036 | 0.58-1.85 | 0.670 | 0.31-1.44 |
| Ordinary level | Ref | -- | Ref | -- | Ref | -- | Ref | -- |
| **Nature of housing** | | | | | | | | |
| Permanent | Ref | -- | Ref | -- | Ref | -- | Ref | -- |
| Semi-permanent | 1.346* | 1.06-1.71 | 1.260 | 0.98-1.63 | 1.093 | 0.65-7.85 | 1.049 | 0.60-1.83 |
| Hut | 1.737 | 0.96-3.15 | 1.306 | 0.67-2.56 | 2.904 | 0.99-8.56 | 1.523 | 0.41-5.70 |
| **Domestic violence witness** | | | | | | | | |
| Yes | 1.331* | 1.03-1.71 | 1.230 | 0.93-1.63 | 1.558 | 0.95-2.56 | 1.476 | 0.84-2.59 |
| No | Ref | -- | Ref | -- | Ref | -- | Ref | -- |
| **Family history of mental illness** | | | | | | | | |
| Yes | 1.321* | 1.05-1.67 | 1.163 | 0.90-1.50 | 2.099** | 1.35-3.27 | 1.994** | 1.22-3.26 |
| No | Ref | -- | Ref | -- | Ref | -- | Ref | -- |
| **Orphan** | | | | | | | | |
| Yes | 1.084 | 0.85-1.39 | 1.028 | 0.79-1.39 | 0.793 | 0.46-1.36 | 0.700 | 0.39-1.27 |
| No | Ref | -- | Ref | -- | Ref | -- | Ref | -- |
| **School sex status** | | | | | | | | |
| Mixed | 1.182 | 0.84-1.67 | 0.935 | 0.60-1.45 | 1.058 | 0.56-2.01 | 0.792 | 0.34-1.86 |
| Single-sex | Ref | -- | Ref | -- | Ref | -- | Ref | -- |
| **School ownership** | | | | | | | | |
| Private | 1.408** | 1.16-1.71 | 1.403** | 1.12-1.76 | 0.972 | 0.66-1.43 | 0.881 | 0.55-1.40 |
| Government | Ref | -- | Ref | -- | Ref | -- | Ref | -- |

Abbreviations: PR=Prevalence Ratio; APR=Adjusted Prevalence Ratio; OR = Odds Ratio; AOR=Adjusted Odds Ratio; Ref=Referent

*<0.05;

**<0.01;

***<0.001

(OR=1.13; 95%CI 1.03 - 1.23) and having a family history of mental illness (OR=2.10; 95%CI 1.35 - 3.27) were associated with having a behavioral disorder. These associations persisted in the adjusted model, as being one year older (adj. OR=1.21; 95%CI 1.05 - 1.38) and having a family history of mental illness (adj. OR=1.99; 95%CI 1.22 - 3.26) had 1.2 and 2.0 times the prevalence of behavioral disorders as compared to younger individuals without mental illness in the family (Table 2). Fig 1 contains a forest plot representation of the multivariable regression models (both logistic and modified Poisson regression models) in Table 2.

## Discussion

This study explored the factors associated with emotional and behavioral disorders among school-going adolescents in Uganda. We revealed two key findings. First, being female and attending private schools were independently associated

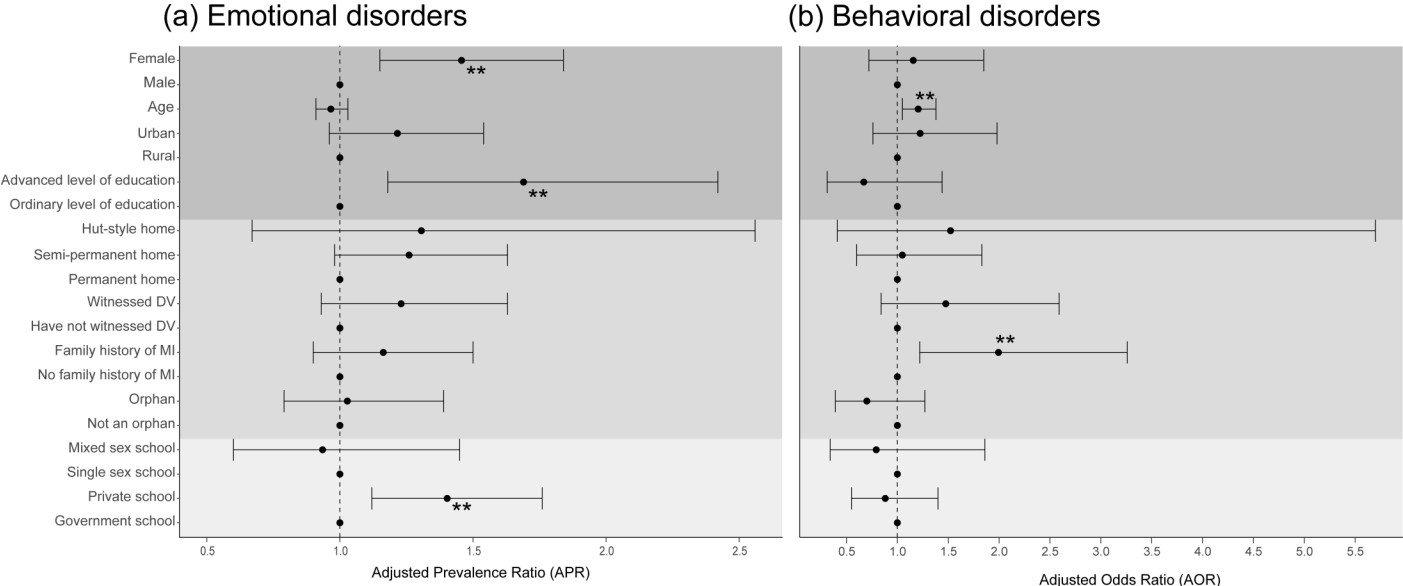

**Fig 1. Logistic regression modeling for (a) emotional and (b) behavioral disorders adjusted for demographic factors (dark grey), hardship experiences (grey), and school features (light grey) among Ugandan school-going adolescents.** *=p<0.05; **=p<0.01; ***=p<0.001.

with emotional disorders. Secondly, older age and having a family history of mental illness were independently associated with behavioral disorders among adolescents attending secondary schools in Central and Eastern Uganda.

The prevalence of emotional disorders (depression, generalized anxiety, social anxiety, and separation anxiety) among this sample of Ugandan adolescents (17.4%) was slightly higher than the values reported in another study from Uganda using the CASI-5 battery (11.5%) [14]. The same study also reported a prevalence of behavioral disorders at 9.6%, slightly higher than the prevalence reported in this study (5.7%). However, the combined prevalence of any emotional or behavioral disorder in this study's sample (20.2%) is consistent with a systematic review of mental disorders in Uganda (22.9%) [31]. The cross-cultural adaptability of the CASI-5 tool has been assessed and documented [24].

The prevalence of psychiatric disorders among school-going adolescents in India, another LMIC, was greater than what was observed in this study, with a significant difference being a higher prevalence of depression (13.7% compared to 0.4% in our study) [32]. A recent systematic review of mental health problems among adolescents in Sub-Saharan Africa reported a similar prevalence of mental health problems (23%), except for a much larger prevalence of depression (19%) [33]. This discrepancy should caution against the generalizability of this study's findings to depression outcomes. By using an alternative approach to categorizing depression in this population (specifically the Symptom Severity T-score approach [23]), we confirmed that this study population had no respondents who fit the threshold for moderate or severe depression as reported in our prior research [30]. We hypothesize that the low prevalence of depression in this population could be a result of multiple factors: the long test battery could have reduced true accuracy of responses (survey fatigue), the prevalence of depression in these schools may be lower than other samples, and the potential response bias with students who are unfamiliar with the PCOs implementing the CASI-5 battery, among other factors. These factors could be contributing to the lower (and potentially misrepresentation of) depression rates in this population.

Our study found that female students were at increased risk for having emotional disorders; however, sex was not significantly associated with having a behavioral disorder. Significant differences in mental and behavioral health outcomes among Ugandan youth according to sex have been found [14,34,35]. Our study revealed twice as many female adolescent students having emotional disorders than their male counterparts. Prior studies consistently reveal that a higher burden of disorders such as depression among females may be linked to biological or socio-cultural experiences [36–38].

Additionally, our study observed that adolescents attending private schools were at increased risk for the presence of an emotional disorder. There is sparse literature investigating this disparity; however, one could hypothesize that increased pressure to achieve academically [39] enhance the emotional burden of adolescents. The presence of caring adults in schools has been shown to influence emotional well-being of school-going adolescents in Sub-Saharan Africa [40].

Our study reported that an adolescent having a family history of mental illness was at increased risk for having a behavioral disorder. While studies have shown that emotional and behavioral disorders can be hereditary (though highly complex in nature) [41,42], family dynamics also play a crucial role in the mental health of adolescents. A variety of interventions have sought to improve family dynamics and address children's mental health challenges, focusing on family strengthening [43], economic empowerment [19], and community support [44]. We did not observe an independent association between family history of mental illness and the presence of an emotional disorder. While family history of mental illness was significantly associated with emotional disorders at the bivariate level, its significance was lost in the multivariable models. This suggests that other confounding factors may play a more significant role in predicting emotional disorders as compared to behavioral.

Some hardship experiences did not emerge as statistically significant risk factors for emotional or behavioral disorders in this population, namely witnessing domestic violence, less-permanent housing, and being orphaned. For some of these factors (nature of housing), the suboptimal statistical power, based on a small number of participants who were among the most vulnerable, may have contributed to the non-significant findings. Regarding the remaining factors, it is possible that underlying confounding factors may be responsible for statistically significant bivariate associations becoming non-significant in our multivariable models.

School-based interventions have become attractive opportunities to improve mental health outcomes among adolescents who attend school [45–47]. Our study findings suggest that further attention is required to focus on Ugandan private secondary schools to enhance mental health promotion activities. A school culture beyond academic responsibility and fostering positive relationships with teachers has been highlighted as potentially crucial for school-based mental health interventions [48]. This study also highlights potential populations to focus interventions on, including, but not limited to, those attending private schools, those with a family history of mental illness, and female students in Uganda. Each of these factors has been shown to independently increase one's risk of having an emotional or behavioral disorder in this study population. Prior research demonstrated that economic and family interventions reduced absenteeism (general and sickness-related) among school-going adolescent girls [49]. However, this intervention did not improve behavior and grade repetition among schooling adolescent girls in Southern Uganda. A recent systematic review identified beneficial school-based interventions to improve mental health; cognitive behavioral behavior therapy and interventions focusing on cultivating a growth mindset were among the most common [50]. Integrating these interventions, or others aiming to reduce school violence [51], could improve the mental health of school-going adolescents in Uganda.

As a result of the cross-sectional study design, this study can only highlight associations with emotional or behavioral disorders; inferences of causality cannot be made with this study's findings alone. The length of the CASI-5 test battery could have introduced potential biases or misrepresentations in this dataset. Respondents answered 202 questions about symptoms they may be experiencing, possible impairment resulting from these symptoms, and demographic details. Suboptimal effort and fatigue while testing is a concern among neuropsychological scientists assessing mental health outcomes in children [52]. Although there was potential for testing fatigue biasing the dataset, we attempted to address this by removing observations where individuals responded to less than half of the survey items. Respondents self-reported symptom frequency and symptom impairment, which may have contributed to slight misrepresentations of true emotional and behavioral health burdens on this population. Major depressive disorder was minimally present within this study population, which differs from other studies; this observation should be explored further to assess whether there are socio-cultural, economic, or environmental explanations beyond tool validity with this population. The rarity of some key independent variable outcomes may have limited this study. For example, very few respondents reported living in huts. In bivariate and adjusted models, living in a hut carried a large effect size, but the rarity of this outcome reduced the findings' statistical confidence. This, however, highlights an area of future investigation.

## Conclusions

Female students, having an advanced level of education, and attending private secondary school are independently associated with the presence of an emotional disorder, while age and having a family history of mental illness were independently associated with having a behavioral disorder. These findings are a call to action for policy makers in the health and education ministries in Uganda and for researchers to focus on improving adolescent emotional and behavioral health outcomes. Additionally, we have highlighted potentially at-risk populations, and identified potential target areas for school-based interventions. Further research is required to assess the causal pathways in school-going adolescents, understand what key experiences may have contributed to poor emotional or behavioral health, and better understand the contributions of less stable housing and other hardship experiences in mental health in this study population.

## Supporting information

**S1 Table. Detailed summary statistics for each emotional and behavioral disorder included in this analysis among Ugandan school-going adolescents.**
(XLSX)

**S1 Data. The deidentified dataset used for the analysis included this study of emotional and behavioral disorders among school-going adolescents in Uganda.**
(DTA)

**S1 Checklist. Inclusivity in global research.**
(DOCX)

## Acknowledgments

We also thank the secondary students' surveys for their support and cooperation.

## Author contributions

**Conceptualization:** Max Bobholz, Catherine Abbo, Laura D. Cassidy, Ronald Anguzu.

**Data curation:** Max Bobholz, Ronald Anguzu.

**Formal analysis:** Max Bobholz.

**Investigation:** Max Bobholz, Catherine Abbo, Ronald Anguzu.

**Methodology:** Max Bobholz, Arthur Kiconco, Ronald Anguzu.

**Project administration:** Catherine Abbo, Laura D. Cassidy.

**Supervision:** Julia Dickson-Gomez, Catherine Abbo, Laura D. Cassidy, Ronald Anguzu.

**Visualization:** Max Bobholz.

**Writing – original draft:** Max Bobholz.

**Writing – review & editing:** Max Bobholz, Julia Dickson-Gomez, Catherine Abbo, Arthur Kiconco, Abdul R. Shour, Simon Kasasa, Laura D. Cassidy, Ronald Anguzu.

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
