## [Decision Letter · Decision Letter 0]

13 Jan 2025

PGPH-D-24-02392

Correlates of behavioral and emotional disorders among school-going adolescents in Uganda

Dear Dr. Abbo,

Thank you for submitting your manuscript to PLOS Global Public Health. After careful consideration, we feel that it has merit but does not fully meet PLOS Global Public Health’s publication criteria as it currently stands. Therefore, we invite you to submit a revised version of the manuscript that addresses the points raised during the review process.

We look forward to receiving your revised manuscript.

Kind regards,

Massimiliano Orri, PhD

Academic Editor

Journal Requirements:

2. Please provide an Author Summary. This should appear in your manuscript between the Abstract (if applicable) and the Introduction, and should be 150–200 words long. The aim should be to make your findings accessible to a wide audience that includes both scientists and non-scientists. Sample summaries can be found on our website under Submission Guidelines:

https://journals.plos.org/globalpublichealth/s/submission-guidelines#loc-parts-of-a-submission.

Additional Editor Comments (if provided):

Reviewers' comments:

Reviewer's Responses to Questions

**Comments to the Author**

1. Does this manuscript meet PLOS Global Public Health’s publication criteria?

Reviewer #1: Yes

Reviewer #2: Yes

2. Has the statistical analysis been performed appropriately and rigorously?

Reviewer #1: Yes

Reviewer #2: Yes

3. Have the authors made all data underlying the findings in their manuscript fully available (please refer to the Data Availability Statement at the start of the manuscript PDF file)?

Reviewer #1: Yes

Reviewer #2: Yes

4. Is the manuscript presented in an intelligible fashion and written in standard English?

Reviewer #1: Yes

Reviewer #2: Yes

Reviewer #1: This is an interesting study on cross-sectional associations between demographic, school characteristics, and hardship, and mental health of adolescents in Uganda. A methodic and efficient approach was used for the sampling during 6 months of 2017, which is well-described in the methods. The sample size seems adequate for the study. The overall aim of the study is to identify at-risk individuals in order to inform intervention strategies for youth mental health in the school setting.

I have a few general comments: I recommend reviewing the structure of the paragraphs to make sure they are linked to one another, using transitions. Most importantly, I recommend a more expansive literature review for both the background and the discussion. More context in which the present study places itself is needed, and more interpretation of the findings in light of prior studies in Uganda or Sub-Saharan Africa is necessary for a more impactful discussion.

Please find more detailed comments per section.

Background

Please add a reference for the second sentence stating a prevalence of 13.9% of mental health disorders in the global population.

Add a reference for the third sentence stating a 13% burden of disease in adolescents.

For the sentence “Mental disorders (…) have risen considerably”, can you provide an idea of the time frame in which this rise has occurred. Also, the reference for this statement is a study from Uganda. It’s unclear whether the statement focusing on LMICs is adequately supported by this reference.

Reference required to support the statement “Depression, anxiety, and behavioral disorders are among (…)”

There is an incoherence in the background from the abstract and the background in the paper. The abstract reports 11.5% of Ugandan adolescents report emotional disorders (which include depression), vs the background in the manuscript reports 23.6% depression prevalence in Ugandan children and adolescents. The abstract does not specify that 11.5% refers to an at-risk population: perinatally HIV-infected youth.

It is not clear what the 5th paragraph transition word “therefore” is linked to. It does not appear to fit with the content of the 4th paragraph, but rather the 3rd one. Please review the structure.

The statement “heterogenous methodologies and outcome assessments complicate this field of study” does not appear to be connected to other points made in the paragraph. I suggest removing or clarifying.

“burden” of emotional and behavioral disorders is used multiple times in the background, could you specify what you mean by “burden”, is it the prevalence of these disorders?

Hardship is a variable of interest, but has not been defined. Could you specify what hardship experiences are included when introducing your study in the last paragraph: “We hypothesized that hardship experiences, demographic characteristics, and school features contribute to and could be associated with the prevalence of emotional and behavioral disorders among school-going adolescents in Uganda.” Same comment for school features, and demographic characteristics.

Methods

The abstract background information does not match the methods information on age range. The abstract states participants are aged 10-24 years. The methods state the participants are aged 10-18 years. I assume it’s the latter. Please correct accordingly.

Results

Participant demographics

The second sentence states: “An overall proportion of 17.36% (n=340) reported symptoms suggestive of a mental health condition, and 5.71% (n=112) had behavioral disorders”. I suppose “mental health condition” refers to emotional disorders. Please correct accordingly.

The sentence “Few respondents (1.48%; n=28) reported living in hut-style, grass-thatched homes, 82.40% (n=1,559) lived in permanent residences, and 16.12% of respondents (n=305) lived in semi-permanent homes (Table 1).” This is the first mention of “grass-thatched homes” which I assume is semi-permanent housing. I suggest to either state earlier that semi-permanent includes grass-thatched homes, or remove grass-thatched in this sentence and replace by “semi-permanent.

Prevalence of behavioral disorders

Prevalence for conduct disorders and oppositional disorders are stated, except for ADHD. I suggest adding the prevalence for all behavioral disorders.

This section includes prevalence of emotional disorders, please modify the structure for the reported results to match your sections.

Please merge this section with “prevalence and correlates of behavioral disorders”

Prevalence and correlates of emotional disorders

Define prevalence ratio then use PR subsequently.

Can you comment on the overlap between emotional and behavioral disorders, did you find participants with both, or are your categories “emotional disorder” and “behavioral disorder” exclusive categories?

Is Figure 1 just an illustration of your results in Table 2, adjusted models? If so, please state.

Please state for what models the logistic vs modified poisson regressions were used in Table 2.

Discussion

Please remove “PTSD” in the second paragraph, it was not included in this study.

In the second paragraph, clearly state the prevalence rates in this study compared to the rates reported in other studies, so we understand the differences in prevalence.

Same comment for the third paragraph, report prevalence rates when stating differences between studies, within country or between countries.

It is unclear why India is mentioned, because it is also LMIC?

Could you elaborate on/provide some interpretation on the discrepancy in depression rates in this study and rates observed in the review on Sub-Saharan Africa?

“By using an alternative approach to categorizing (…)” seems to refer to a supplementary analysis done by the authors but has not been previously described. Could this be integrated earlier in the paper?

Could you expand on why emotional disorders may be higher in private schools? Have there been other studies in Uganda linking private school to stress, or mental health, compared to public schools?

Please include a discussion on the association between family history of mental illness and behavioral disorders.

Please include a discussion on variables which were not associated to emotional and behavioral disorders (e.g. domestic violence, orphan, urbanity, nature of housing).

Could you give concrete recommendations for interventions in the school-setting, according to prior literature? For example, there is literature associating school violence to mental health in Uganda, specifically in girls (Devries, K. M., Child, J. C., Allen, E., Walakira, E., Parkes, J., & Naker, D. (2014). School violence, mental health, and educational performance in Uganda. Pediatrics, 133(1), e129-e137). Are there other factors which may explain the associations that you find?

Reviewer #2: Comments.

1.Background

It is better to put statement in paragraph two in the place of paragraph one; please revise this section to make clear understanding for your study.

2.Method

Who are respondents for this study? - Adolescents or their parents or both of them. Please make it clear and describe in details to reduce bias. Collecting data from adolescents or from their parents raise the risk of recall bias; so you should be explain what you did to reduce information bias from respondents in limitation section of your study.

3.Conclusion

Recommendation should be based on finding, you can recommend for factors associated with outcome variable in your study. Make it smart since conclusion is final output for your study.

Final what you did for the respondents who had emotional and behavioral problems in your study. Please specify your intervention for them if did for them.

4.Figures not visible, revise it

5.Check the references for clearly state author, journal name, volume, page number, etc.

**Do you want your identity to be public for this peer review?** For information about this choice, including consent withdrawal, please see our Privacy Policy

Reviewer #1: No

Reviewer #2: **Yes: ** Tamene Berhanu Alaho

---

## [Decision Letter · Decision Letter 1]

1 Apr 2025

PGPH-D-24-02392R1

Correlates of behavioral and emotional disorders among school-going adolescents in Uganda

Dear Dr. Abbo,

Thank you for submitting your manuscript to PLOS Global Public Health. After careful consideration, we feel that it has merit but does not fully meet PLOS Global Public Health’s publication criteria as it currently stands. Therefore, we invite you to submit a revised version of the manuscript that addresses the points raised during the review process.

We look forward to receiving your revised manuscript.

Kind regards,

Massimiliano Orri, PhD

Academic Editor

Journal Requirements:

Additional Editor Comments (if provided):

Reviewers' comments:

Reviewer's Responses to Questions

**Comments to the Author**

Reviewer #1: (No Response)

publication criteria?

Reviewer #1: (No Response)

3. Has the statistical analysis been performed appropriately and rigorously?

Reviewer #1: (No Response)

4. Have the authors made all data underlying the findings in their manuscript fully available (please refer to the Data Availability Statement at the start of the manuscript PDF file)?

Reviewer #1: (No Response)

5. Is the manuscript presented in an intelligible fashion and written in standard English?

Reviewer #1: (No Response)

Reviewer #1: I thank the authors for their responses on the comments of the first round of revisions. However, some issues particularly related to the discussion remain.

First, I'd like to comment on the author's response: the pages and line numbers reported in the author responses did not correspond to the lines and pages of the revised manuscript. This does not facilitate the review process.

INTRODUCTION

The 4th paragraph of the introduction is not connected to the previous or the next paragraphs (3rd and 5th). Although the content of the paragraph is important because it introduces the outcome measure for this study, which is reported to be validated in Uganda, the relevance of the content is not stated. I strongly suggest adding transition sentences between paragraphs in general. The flow of ideas (how each paragraph leads to the next) is not clear.

RESULTS

Reiterating my previous comment: results on emotional disorders are presented in the behavioral disorders section of your results.

DISCUSSION

In the discussion, you list emotional disorders "The prevalence of emotional disorders (depression, generalized anxiety, ADHD, social anxiety, and separation anxiety)", however ADHD is in your "behavioral disorders" in your results section. Remove ADHD.

Please add a hypothesis as to why your study's depression prevalence rates are so low (0.4%) compared to other samples. In your response you mention response fatigue, survey design; how would this contribute to reporting low depressive symptoms? Can you add a hypothesis in your discussion as to why your rates may be so low.

In the paragraph on family history of mental health predicting behavioral disorders, please expand on why this association was found, but not with emotional disorders. You mention how family dynamics may play a role in youth mental health, expand on how that is linked to mental illness in the family and youth mental health, specifically behavioral disorders.

**Do you want your identity to be public for this peer review?** For information about this choice, including consent withdrawal, please see our Privacy Policy

Reviewer #1: No

---

## [Editor Report · Decision Letter 2]

2 May 2025

Correlates of behavioral and emotional disorders among school-going adolescents in Uganda

PGPH-D-24-02392R2

Dear Assoc Professor Abbo,

We are pleased to inform you that your manuscript 'Correlates of behavioral and emotional disorders among school-going adolescents in Uganda' has been provisionally accepted for publication in PLOS Global Public Health.

Best regards,

Massimiliano Orri, PhD

Academic Editor